# Human microbiome variation associated with race and ethnicity emerges as early as 3 months of age

Elizabeth K. Mallott[1,2,3]*, Alexandra R. Sitarik[4], Leslie D. Leve[5], Camille Cioffi[5], Carlos A. Camargo, Jr[6], Kohei Hasegawa[6], Seth R. Bordenstein[1,2,7,8,9,10,11]*

1 Vanderbilt Microbiome Innovation Center, Vanderbilt University, Nashville, Tennessee, United States of America, 2 Department of Biological Sciences, Vanderbilt University, Nashville, Tennessee, United States of America, 3 Department of Biology, Washington University in St. Louis, St. Louis, Missouri, United States of America, 4 Department of Public Health Sciences, Henry Ford Health, Detroit, Michigan, United States of America, 5 Prevention Science Institute, University of Oregon, Eugene, Oregon, United States of America, 6 Department of Emergency Medicine, Massachusetts General Hospital, Harvard Medical School, Boston, Massachusetts, United States of America, 7 Vanderbilt Genetics Institute, Vanderbilt University Medical Center, Nashville, Tennessee, United States of America, 8 Vanderbilt Institute for Infection, Immunology, and Inflammation, Vanderbilt University Medical Center, Nashville, Tennessee, United States of America, 9 Department of Pathology, Microbiology, and Immunology, Vanderbilt University Medical Center, School of Medicine, Nashville, Tennessee, United States of America, 10 Departments of Biology and Entomology, Pennsylvania State University, University Park, Pennsylvania, United States of America, 11 The One Health Microbiome Center, Huck Institutes of the Life Sciences, Pennsylvania State University, University Park, Pennsylvania, United States of America

* mallott@wustl.edu (EKM); s.bordenstein@psu.edu (SRB)

**Data Availability Statement:** Sequencing data and metadata included in this study was downloaded from NCBI's Sequence Read Archive (accessions:

## Abstract

Human microbiome variation is linked to the incidence, prevalence, and mortality of many diseases and associates with race and ethnicity in the United States. However, the age at which microbiome variability emerges between these groups remains a central gap in knowledge. Here, we identify that gut microbiome variation associated with race and ethnicity arises after 3 months of age and persists through childhood. One-third of the bacterial taxa that vary across caregiver-identified racial categories in children are taxa reported to also vary between adults. Machine learning modeling of childhood microbiomes from 8 cohort studies (2,756 samples from 729 children) distinguishes racial and ethnic categories with 87% accuracy. Importantly, predictive genera are also among the top 30 most important taxa when childhood microbiomes are used to predict adult self-identified race and ethnicity. Our results highlight a critical developmental window at or shortly after 3 months of age when social and environmental factors drive race and ethnicity-associated microbiome variation and may contribute to adult health and health disparities.

## Introduction

Two major goals of the human microbiome sciences include increasing the representation of undersampled groups in microbiome datasets [1–3] and understanding the tempo by which inequitable experiences, intergenerational inequality, and structural racism impact

PRJNA322554, PRJEB11697, and PRJEB13896), QIITA (studies 11129 and 10894), and FigShare (https://doi.org/10.6084/m9.figshare.7011272.v3). Additional sequencing data and metadata for included studies are available as outlined in the original publications, which are listed in S1 Table. All data necessary to reproduce main text and supplementary figures are included in S1–S15 Data files. Code for all analyses can be found on GitHub (https://github.com/BordensteinLaboratory/Childhood_micro_metaanalysis) and are archived on Zenodo (https://doi.org/10.5281/zenodo.8063024).

**Funding:** The WHEALS study was supported by P01 AI089473-01 from the National Institutes of Health (Bethesda, MD) (ARS). The MARC-35 study was supported by UG3/UH3 OD-023253 from the National Institutes of Health (Bethesda, MD) (CAC and KH). The Early Growth and Development Study microbiome data was supported by UH3 OD023389, P50GM098911, R01 DA035062 from the National Institutes of Health (Bethesda, MD), and a Faculty Alumni Award from the College of Education, University of Oregon (LDL and CC). EKM was supported by the Vanderbilt Microbiome Innovation Center. SRB was supported by the One Health Microbiome Center in the Huck Institutes at The Pennsylvania State University. The funders had no role in study design, data collection and analysis, decision to publish, or preparation of the manuscript.

**Competing interests:** The authors have declared that no competing interests exist.

**Abbreviations:** AGP, American Gut Project; ANCOM-BC, analysis of compositions of microbiomes with bias correction; ASV, amplicon sequence variant; AUC, area under the curve; IBS, irritable bowel syndrome; PERMANOVA, permutational multivariate analysis of variance; prAUC, precision recall AUC; SES, socioeconomic status.

microbiome variation and health outcomes [4–8]. Early-life social and environmental exposures can have large and lasting effects on child development and adult health, and perturbations to the gut microbiome may be important to future disease risk [9–19]. In the United States, adult gut microbiome diversity correlates with self-identified race and ethnicity [1,3]. However, socioeconomic status (SES)—neighborhood deprivation index, individual and parental education, or household income—is both correlated with adult gut microbiome diversity and is associated with race and ethnicity [20–24]. We emphasize that race and ethnicity are proxies for inequitable exposure to social and environmental determinants of health due to structural racism [6–8,25,26]. When human microbiome differences arise during development and whether or not distinguishing gut taxa overlap between childhood and adulthood are key questions that have implications for long-term effects of early life experiences, including structural racism, on microbiome variation.

To identify the developmental window when microbiome variation emerges, how long it persists during childhood, and which distinguishing taxa overlap between children and adults, we combined 8 gut microbiome composition datasets from 2,756 samples spanning 729 children between birth and 12 years of age throughout the US (S1 Table). We used caregiver-identified race (Asian/Pacific Islander, Black, White) and ethnicity (Hispanic, non-Hispanic) to capture complex interactions of multiple biosocial factors that influence gut microbiome composition, even though race and ethnicity are not biological categories that directly influence microbiome variation [5–7,26]. We used a diverse dataset of childhood microbiome samples to identify features of the gut microbiome that are potential markers of the inequitable experiences underlying health disparities. We selected samples from multiple 16S rRNA gene sequencing studies that represent a higher diversity of children than is commonly present in large analyses of the gut microbiome [1–3]. In the present study, 17.2% of samples were from non-White individuals, and 14.3% of samples were from Hispanic individuals. While the majority of samples from Hispanic individuals are from Hispanic White children, some Hispanic Black children are present in the dataset.

## Results

### Microbiome variation emerges at or shortly after 3 months of age

Subject explained the greatest proportion of variation, consistent with other studies of the gut microbiome (S1 Fig). As age had the second strongest association with gut microbiome composition of the variables tested (Figs 1 and S1–S9 and S2–S4 Tables), we stratified samples by age and analyzed each age category separately while controlling for study differences to disentangle when in development race and ethnicity-associated microbiome variation originates. Delivery route and infant diet were not included in the age-stratified analysis, as they covaried with race and ethnicity (S10 and S11 Figs and S5 Table).

Notably, race and ethnicity did not significantly vary with gut microbiome alpha diversity (within-individual diversity) or beta diversity (between-individual diversity) in the early weeks and months of life, including the first week, 1 to 5.9 weeks, and 6 weeks to 2.9 months (permutational multivariate analysis of variance (PERMANOVA), all $p > 0.05$) (Figs 2, S2, S12, and S13 and S2 Table). However, at 3 to 11.9 and 12 to 35.9 months, gut microbiome composition based on UniFrac distances varied slightly but significantly by both race and ethnicity (PERMANOVA, all $p < 0.05$) (Figs 2B, S2, S12, and S13 and S2 Table). Additionally, most measures of alpha diversity varied across racial categories at 3 to 11.9 months and across both racial and ethnic categories at 12 to 35.9 months (LME, $p < 0.05$) (Fig 2A and S4 Table). Pairwise comparisons confirmed that Black individuals had higher within-sample diversity than White individuals at 3 to 11.9 and 12 to 35.9 months for at least one of the 5 measures of diversity (Fig 2A

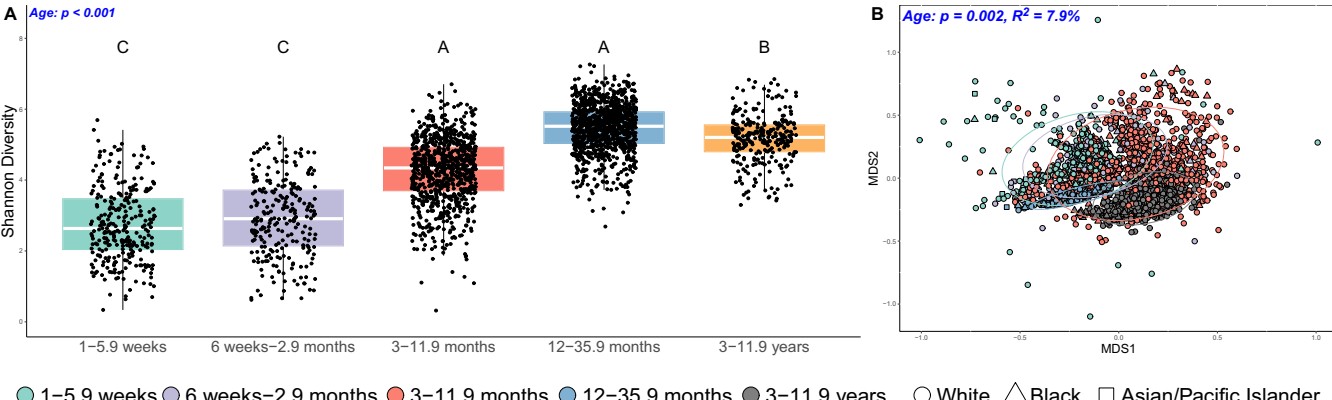

**Fig 1. Age structures variation in the gut microbiome.** (**A**) Boxplots show increases in Shannon diversity with age, and (**B**) nonmetric multidimensional scaling (NMDS) plots show a significant association of age with weighted UniFrac distances. Colors and 95% confidence ellipses denote age, and shape denotes race. Blue text in the panels highlights significant *p*-values. Data underlying this figure can be found in S1 and S2 Data.

and S4 Table) [27]. While higher alpha diversity is consistently associated with better cardio-metabolic health and lower incidence of inflammatory disease in adults [28–30], studies have found mixed results in children. For example, studies of associations between alpha diversity and risk of allergic disease have found negative [31], positive [32], and no [33] association. From 3 to 11.9 years, race associated with gut microbiome composition using only unweighted UniFrac distances (PERMANOVA, all $p < 0.05$) (S12 and S13 Figs and S2 Table). Collectively, these results reveal that race and ethnicity associate with microbial diversity after 3 months of age, and, notably, this variation persists through childhood years.

## Child gut microbiome variation recapitulates that of adults

To identify differentially abundant taxa, we used analysis of compositions of microbiomes with bias correction (ANCOM-BC) for each variable of interest across all age categories. Age was included as a factor in the models, and numerous taxa were differentially abundant across age categories (S6–S9 **Tables**). The abundances of several taxa significantly were associated with race and/or ethnicity in all samples combined (S5–S9 **Tables**), including several that varied in abundance between age categories (S14 and S15 Figs). Taxa positively associated with breastfeeding (*Bifidobacterium*, *Lactobacillus*, and *Staphylococcus*) [34,35] were significantly negatively correlated with age, as expected (S14 and S15 Figs and S9 Table). These taxa were differentially abundant between racial or ethnic categories, likely due to differences in rates of breastfeeding across these groups (S10 and S11 Figs and S5 Table). Delivery route also differed between racial and ethnic categories—vaginal delivery was more likely than expected in White, Asian/Pacific Islander, and non-Hispanic children and less likely than expected in Black and Hispanic children (S10 and S11 Figs and S5 Table). However, some individual species within *Bacteroides*, which is often more abundant in vaginally delivered children [34,35], were more enriched in Black and Hispanic children (S9 Table), contrary to our expectations.

Notably, there was moderate overlap between studies for differentially abundant taxa (S10 Table). Of the 57 gut microbial taxa that varied in abundance between children of differing self-identified racial categories, 19 were previously identified as differentially abundant between Black and White adult individuals in a recent controlled study of gut microbiome variation [3] (Fig 3A and S9 Table). Four of the 19 overlapping taxa were higher in abundance in both Black children and adults compared with White children and adults, and 4 of the

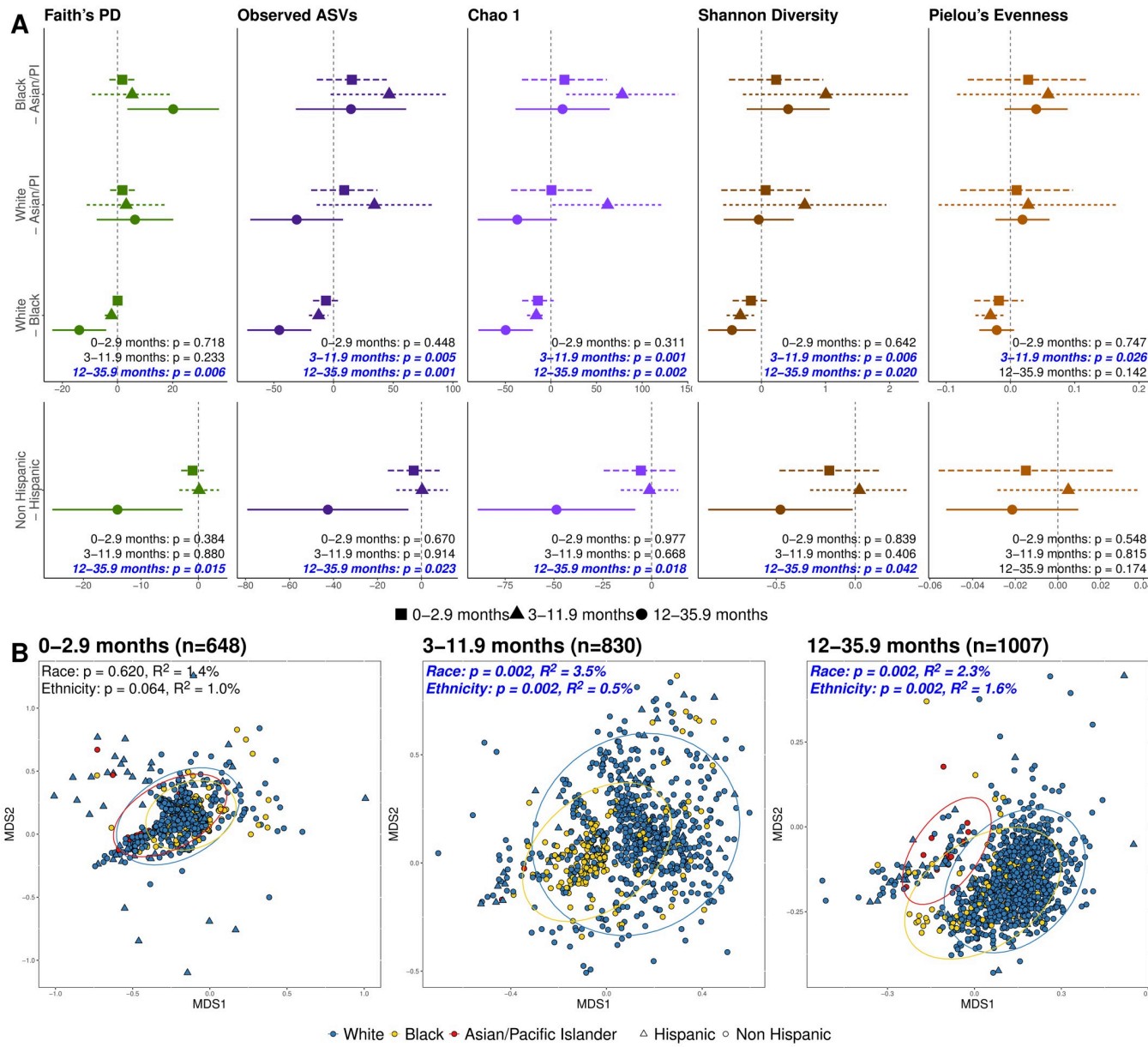

**Fig 2. Microbiome variation emerges at or shortly after 3 months of age.** (**A**) Dot and whisker plots show estimates for Tukey pairwise comparisons in the alpha diversity linear mixed effects models. Dots indicate the estimated difference in alpha diversity when accounting for other covariates in the model, whiskers denote 95% confidence intervals, and the dashed line indicates zero or no difference. Comparisons with whiskers that do not cross zero indicate a significant difference in alpha diversity between those 2 categories. Colors in the dot whisker plots denote alpha diversity metric, and dot shape and line type denote age category. (**B**) NMDS plots show weighted UniFrac distances between by race and ethnicity at 0–2.9 months, 3–11.9 months, and 12–35.9 months. Colors and 95% confidence ellipses in the NMDS plots denote race, and shape denotes ethnicity. Blue text in the panels highlights significant *p*-values. NMDS plots for additional age categories and unweighted UniFrac distances can be found in the Supporting information (S12 and S13 Figs). Data underlying this figure can be found in S1, S2, and S4 Data.

overlapping taxa were lower in abundance in both Black children and adults. The remaining 11 overlapping taxa were either differentially abundant between either Asian/Pacific Islander children and Black children or Asian/Pacific Islander children and White children, or the direction of effect differed between Black and White adults and children. Among the 8 taxa that overlapped and had the same effect in children and adults, *Haemophilus* spp. and

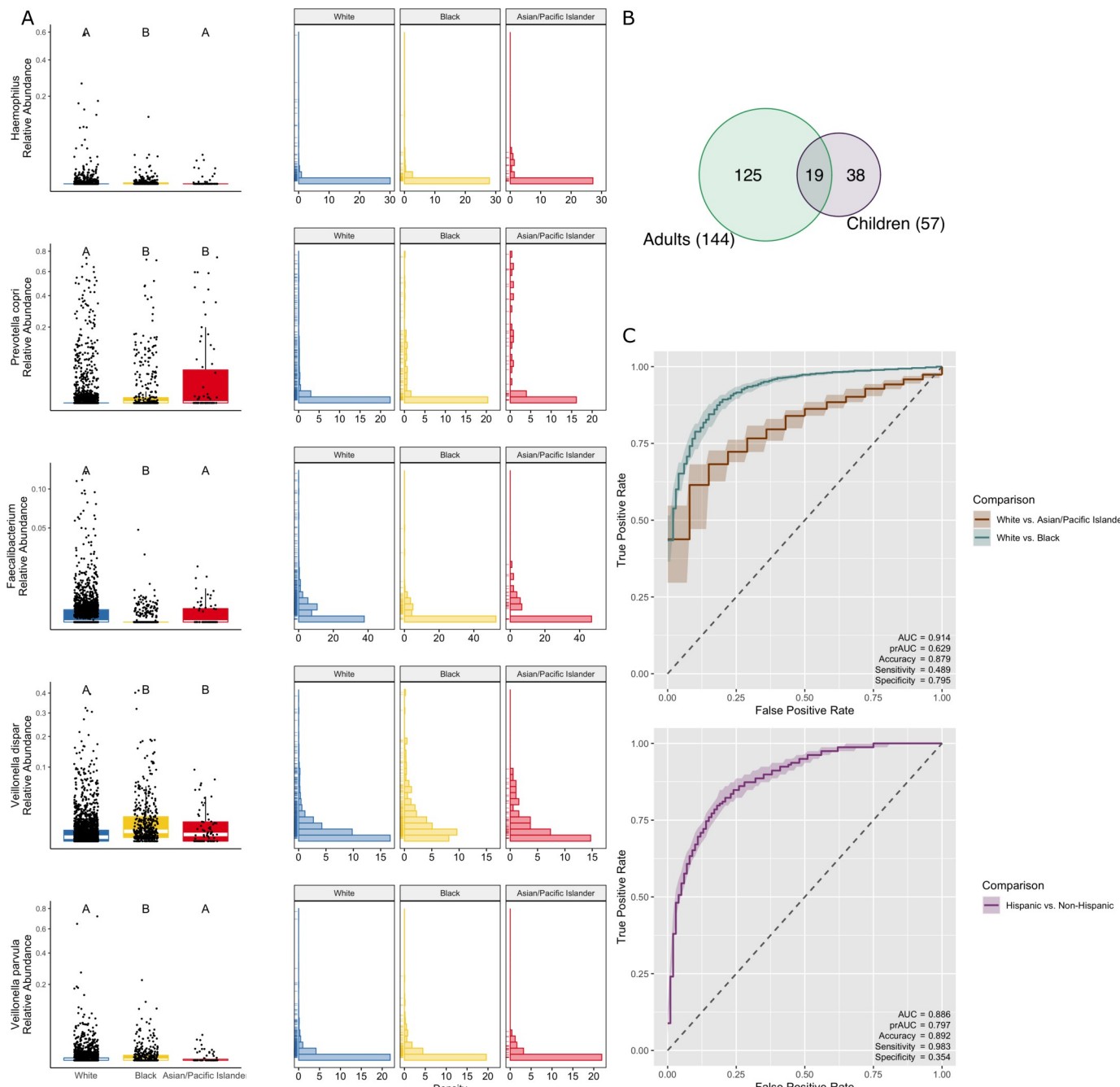

**Fig 3. Child gut microbiome variation recapitulates that of adults.** (**A**) Boxplots showing the relative abundance of select taxa identified as differentially abundant using ANCOM in the current study that overlap with taxa identified as differentially abundant in adults [3]. All boxplots show the median and interquartile range (IQR), and whiskers extend to 1.5*IQR. Relative abundances for boxplots and histograms are square root transformed. (**B**) Venn diagram showing overlapping taxa that are differentially abundant in the gut microbiome between Black individuals and White individuals in the present study in children and in previously published work in adults. (**C**) Receiver operating characteristic (ROC) curves for a random forest model classifying race and ethnicity metadata based on the gut microbiome. Shading represents a 50% confidence interval around the median. Overall model accuracy for race and ethnicity was >87% (the percentage of samples correctly classified as Asian/Pacific Islander, Black, or White and Hispanic and non-Hispanic). Data underlying this figure can be found in S5, S6, and S7 Data and S9 Table.

*Prevotella copri* are higher in abundance in both Black children and adults compared to White individuals (*Haemophilus* spp.: log2 fold change $(log2FC)_{adults}$ = 0.712, $log2FC_{children}$ = 0.739; *P. copri*: $log2FC_{adults}$ = 5.110, $log2FC_{children}$ = 2.513) (ANCOM-BC, all q < 0.05) (Fig 3C). These taxa have been associated with an increased risk of autoimmune and allergic diseases, asthma, and obesity across humans in Europe and North America [28,36–39]. *Faecalibacterium*, which is generally considered to be protective against inflammation [33,40], is lower in abundance in Black children and adults compared to White individuals ($log2FC_{adults}$ = −1.356, $log2FC_{children}$ = −0.230) (ANCOM-BC, q < 0.05) (Fig 3C). Conversely, *Veillonella*, which is associated with a decreased risk of asthma and allergic disease [33,36], is consistently lower in abundance in White children (*Veillonella dispar*: $log2FC_{adults}$ = 1.295, $log2FC_{children}$ = 0.550; *Veillonella parvula*: $log2FC_{adults}$ = 3.321, $log2FC_{children}$ = 1.010) (ANCOM-BC, both q < 0.05) (Fig 3C). Thus, we are finding higher relative abundances of at least one taxon that is positively associated with health in Asian/Pacific Islander, Black, and White children, highlighting the complexity of linking the relative abundance of individual gut microbial taxa to health as a whole. We do note, however, that several of the 19 differentially abundant taxa that overlap between adults and children (S8 Table) have also been found to be associated with SES and unfavorable social and environmental exposures [10,23,41,42].

To detect differentially abundant taxa within each age category, we used generalized linear mixed models with a negative binomial distribution (ANCOM-BC requires more samples per group than we had within each age category). However, few taxa were identified as differentially abundant within each age category (S6–S9 Tables). No phyla or families were differentially abundant between racial and ethnic categories within any age category, and only one genus differed between White and Asian/Pacific Islander children (S6–S9 Tables). Of the 6 species that differed in abundance between racial categories and 4 species that differed in abundance between ethnic categories, none were found in more than one age group (S9 Table). *Coprococcus*, one of the differentially abundant taxa within a specific age group (12 to 35.9 months), was more abundant in non-Hispanic children and has been previously associated both with obesity and a high-fiber diet [43]. The other differentially abundant taxa within specific age groups did not have clear links to health-related outcomes in the literature. Overall, taxa with age-associated variation did not systematically vary by race or ethnicity.

We next used a machine learning approach to identify additional characteristics of the microbiome that may be markers of inequitable exposure to social and environmental determinants of health. A random forest classifier based on the abundance of genera spanning all childhood samples distinguished Black versus White versus Asian/Pacific Islander categories and Hispanic versus non-Hispanic categories with 87% accuracy. Notably, 13 amplicon sequence variants (ASVs) among the top 30 most important genera that increased classification accuracy in the model (S16 and S17 Figs and S11 Table) are taxa identified as differentially abundant between self-identified racial categories in both children in the current study and adults in previous work [3] (Fig 3B and S9 Table). For race, we used a 3-part model, and model performance estimated as area under the curve (AUC; values above 0.5 indicate the classifier is performing better than chance) was 0.914 (Fig 3B). For ethnicity, we used a binary model, and AUC was 0.886 (Fig 3B).

Additionally, we used the childhood microbiome data in a random forest model to assess if childhood microbiome variation predicts that of healthy adults in the American Gut Project (AGP) dataset. As expected, compositional data from children did not reliably distinguish adults of differing racial categories (S18 and S19 Figs), with an AUC of 0.570. Twenty-six of the top 30 taxa identified as important microbiome characteristics in the model using data from children to predict adult metadata were also identified as important taxa in the random forest model that only used data from children (S16 and S19 Figs). However, the taxa with the

highest importance differed with respect to the magnitude and direction of the differences between adults and children (S20 Fig).

Specifically, Enterobacteriaceae and *Prevotella* are highly important in child–child models but are of modest importance in child–adult models (S16 and S19 Figs), and their relative abundances are lowest in White children but highest in White adults (S20 Fig). Other studies have similarly found that specific taxa can be used to differentiate the gut microbiome of groups of people but that the direction of effect can differ between adults and children. *Prevotella* was highly important in both adult and child random forest models used to detect taxa that distinguish the gut microbiome across geographic regions, but the direction of the differences in relative abundance differed [44]. In children, *Prevotella* was more abundant in the US, but *Prevotella* was more abundant in adults outside of the US [44]. *Alistipes* was found to be protective against irritable bowel syndrome (IBS) in adults, but predictive of IBS in children [45].

In contrast, other taxa have a similar direction of effect in both children and adults. *Ruminococcus* is specifically important in the child–adult models, likely due to similar variation in abundance between racial categories in both children and adults (S20 Fig). Higher abundances of *Ruminococcus* are linked with an increased risk of colorectal cancer [46], a disease for which there is a known racial health disparity [47,48]; however, we find that *Ruminococcus* is most abundant in White individuals, a group whose colorectal cancer risk is lower than that of Black individuals but higher than that of Asian/Pacific Islander individuals. Race-associated variation in the relative abundance of *Ruminococcus* across adult guts is not universal, is likely due to a subset of *Ruminococcus* species, and may interact with other factors such as stress or BMI [1,49]. Thus, it is difficult to know how or if the differences observed in the microbiome here contribute directly to health disparities.

## Discussion

Race and ethnicity associate with gut microbiome composition and diversity beginning at 3 months of age, indicative of a narrow window of time (at or shortly after 3 months) and tempo when this variation emerges. Specifically, we found both race and ethnicity account for small but statistically significant proportions of the variation in gut microbiome composition, multiple taxa were differentially abundant between self-reported racial and ethnic categories, several of which were previously identified as differentially abundant in adults [3], and a random forest classifier reliably distinguishes caregiver-identified race and ethnicity. Notably, our findings do not support race- or ethnicity-associated variation appearing at birth or shortly after, when mother-to-infant and other mechanisms of vertical microbial transmission are expected to be strongest [50,51]. None of the differentially abundant taxa identified in the current study are known to be vaginally acquired by infants, and only 2 species are known to be vertically transmitted from the mother [51]. Instead, external factors are most likely shaping race- and ethnicity-associated microbiome variation at or shortly after 3 months. Our results highlight the impetus to increase the diversity of individuals included in studies in the microbiome sciences [1–3] and support the call for studies investigating how structural racism and other structural inequities affect microbiome variation and health [4–7].

The race- and ethnicity-associated differences in the gut microbiome likely reflect differences in environmental and social factors [6–8,25,26]. In the US, there are clear racial and ethnic disparities in health that are tied to differences in these same factors—psychosocial stressors, socioeconomic differences, culture, diet and access to food, access to healthcare and education, interactions with the built environment, and environmental pollutants [6,25,49,52,53]. These factors are important social and environmental determinants of health that have tangible impacts through the modification of human physiology [52,53]. In addition,

there is evidence that the developmental trajectory of the gut microbiome is associated with immune system development, metabolic programming, antibiotic resistance, and risk of asthma, allergic, and autoimmune disease [17,33,36,54–60]. Thus, variation in social and environmental determinants of health that is associated with race and ethnicity may not only shape microbiome variation and impact health but also contribute to health disparities [6,7,20,25]. The tempo and types of factors contributing significantly to race- and ethnicity-associated gut microbiome variation are a priority for research.

Previous studies have identified race- and ethnicity-associated variation in the gut microbiome of children [27,61–64], though they did not pinpoint when in development variation appears and the association is not consistent across studies [36,41,65–73]. In particular, previous work demonstrated that sociodemographic factors related to rates of exposure to stress, access to grocery stores and healthcare, and environmental exposure risk are correlated with race-associated variation in the gut microbiome and that the effect of some of these factors, such as household income, are stronger in infants compared with neonates [27]. Due to the limitations of available metadata for all studies, we were not able to include all factors known to be important in our analysis, such as antibiotic exposure [10,27,74,75], environmental microbial exposures [27,34,56,76], childhood diet [54,70], and various measures of maternal health during pregnancy [9,27,54,63,66,72,77–79]. Many of the studies did not measure potentially important factors that are associated with race and ethnicity, including SES, discrimination or stress, and detrimental environmental exposures. Factors that are known to impact gut microbiome composition and were included in our models—age, sex, delivery route, and infant diet—were not independent of race and/or ethnicity (S10 and S11 Figs and S5 Table). While our study included a relatively high proportions of non-Hispanic Black and Hispanic White children, our inferences were limited by low numbers of Asian American/Pacific Islander children. The datasets used in the current study did not have a sufficient number of Middle Eastern, Native American, and Alaskan Native children to include those individuals in the analysis.

Self-identified race and ethnicity are complex concepts and have limitations. Self-identification varies over time, may not be reflected by predetermined categories used in surveys, and may not capture all aspects of race and ethnicity [80–82]. An additional limitation is that the majority of included studies were conducted in urban areas in distinct geographic locations. The data may not be representative of children from rural areas or the entirety of the US. The results of our study are also not generalizable to other countries due to cultural variation in definitions of racial and ethnic categories. These limitations highlight the necessity of future efforts to recruit a far greater diversity of participants for understanding human microbiome diversity [1–3].

During the first 3 months of age, typically high inter- and intraindividual variability in the infant gut microbiome may contribute to the effect of race and ethnicity, in addition to other maternal, environmental, and social factors that associate with the gut microbiome during this developmental period [35,83,84]. Additionally, the rapid development and marked variation in abundance of microbial taxa within and between individuals continues for at least the first year of life [34,85,86]. Differences in social exposures through childcare, dietary variation due to differential rates of breastfeeding and methods of starting solid food, and environmental exposures through time spent in green spaces may be especially impactful starting at 3 months of age and continuing throughout the first year [9–19,87,88]. Many studies of early life and external factor associations with gut microbiome variation have had limited power to detect the effects of multiple factors, finding few or inconsistent relationships between early life determinants and gut microbiome diversity and composition [10,17,76]. Our findings underscore the need for well-powered, longitudinal studies of diverse cohorts that comprehensively assess

all internal and external factors known to affect the developmental trajectory of the microbiome [5–7,25,89–92]. Other studies have found that the development of the gut microbiome appears to be particularly sensitive to environmental factors and early life events during the first 3 years of life [14,34,93,94]. Additional work is now needed to assess if social and environmental determinants of health begin to influence variation in the microbiome at or near 3 months of age in a way that is potentially important for understanding health disparities in adults, providing a relatively narrow window of time in which to identify potentially impactful factors.

## Materials and methods

Eight datasets with 16S rRNA sequencing data and available race and ethnicity metadata were used in this study [27,66,67,70,72,95,96] (S21 Fig and S1 Table). Individuals between birth and 12 years of age, living in the US, with a caregiver-reported race of Black, White, or Asian/Pacific Islander, and with a caregiver-reported ethnicity of Hispanic or non-Hispanic were included in the analysis. Individuals were not selected based on a known disease phenotype (e.g., type 1 diabetes). Study was included in all models as strata to control for the effects of different study parameters, and individual identity was included as a factor in all models to assess the impact of individual differences on microbiome communities. While sequencing method, primer choice, and sequencing depth did have a significant association with microbial community composition when included in models, including study as strata removed the effect of these study-specific parameters (S2 Table). As some of the included studies had multiple participants from the same family, we also tested if individual identity or family had a larger effect size. In all cases, individual identity explained a larger proportion of the variation than family (S2 Table).

Sequence analyses were carried out in QIIME2 (v.2021.4) [97]. Each study was individually imported into QIIME, and the DADA2 algorithm was used to denoise each study separately to allow us to use appropriate trimming and truncation parameters for each dataset. Feature tables and representative sequences from all studies were then merged using the fragment insertion method [98] to control for differences in amplification and sequencing methodologies between studies. The merged table was filtered to remove sequences absent from the insertion tree. Taxonomy was assigned using a Naïve-Bayesian classifier trained on the Greengenes 13_8 99% OTU full-length 16S rRNA gene sequence database. Mitochondria and chloroplast sequences were filtered from the merged feature table prior to downstream analysis.

Alpha and beta diversity indices were calculated in QIIME and exported for statistical analysis in R [99]. Linear mixed effects models as implemented in the *lme4* package [100] were used to detect significant associations between race, ethnicity, age, sex, delivery route, and infant diet on multiple measures of within-sample diversity (Faith's PD, observed ASVs, Chao 1, Shannon diversity, and Pielou's evenness). Study and individual identity were included as random effects in all linear models to control for the effects of different study parameters and repeatedly sampling individuals. PERMANOVA, as implemented in the *vegan* package [101], was used to examine associations between race, ethnicity, age, sex, delivery route, and infant diet on unweighted and weighted UniFrac distances (example model: WeightedUniFrac ~ Race + Ethnicity + Age + Sex + Delivery route + Infant diet + SubjectID, strata = Study). Study was included as the strata in the PERMANOVA models to constrain permutations within each study and control for study-specific methodological differences in sample collection and processing. For both the alpha and beta diversity analyses, we additionally examined the effect of sequencing technology, primer set, and sequencing depth (S2 and S4 Tables) (S7–S9 and S21 Figs). Analysis of composition of microbiomes was used to identify differentially abundant

phyla, families, genera, and species across all samples using the *ANCOM-BC* package [102]. Generalized linear models using a negative binomial distribution were used to detect differentially abundant phyla, families, genera, and species within each age category using the *glmmTMB* package [103]. Random forest classification was performed using the *mikropml* package [104] in R. A total of 100 training/test data splits were used for each model, and 5-fold cross-validation was repeated 100 times for each of the 100 training/test data splits using the default settings of the *run_ml()* command. Median AUC, precision recall AUC (prAUC), accuracy, sensitivity, and specificity are reported for each model.

## Supporting information

**S1 Text. Impact of age, delivery mode, and infant diet on gut microbiome composition and diversity.**
(DOCX)

**S1 Fig.** Nonmetric multidimensional scaling plots showing the effect of race on weighted (**A**) and unweighted (**B**) UniFrac distances in all samples combined. Data underlying this figure can be found in S2 and S3 Data.
(TIF)

**S2 Fig. Nonmetric multidimensional scaling plots showing the effect of age on unweighted UniFrac distances.** Data underlying this figure can be found in S2 and S3 Data.
(EPS)

**S3 Fig.** Nonmetric multidimensional scaling plots showing the effect of sex on weighted (**A**) and unweighted (**B**) UniFrac distances. Data underlying this figure can be found in S2 and S3 Data.
(TIF)

**S4 Fig.** Nonmetric multidimensional scaling plots showing the effect of delivery mode on weighted (**A**) and unweighted (**B**) UniFrac distances. Data underlying this figure can be found in S2 and S3 Data.
(TIF)

**S5 Fig. Nonmetric multidimensional scaling plots showing the effect of infant diet on unweighted and weighted UniFrac distances.** Data underlying this figure can be found in S2 and S3 Data.
(TIF)

**S6 Fig. Nonmetric multidimensional scaling plots showing the effect of study on unweighted and weighted UniFrac distances.** Data underlying this figure can be found in S2 and S3 Data.
(TIF)

**S7 Fig. Nonmetric multidimensional scaling plots showing the effect of sequencing technology on unweighted and weighted UniFrac distances.** Data underlying this figure can be found in S2 and S3 Data.
(TIF)

**S8 Fig. Nonmetric multidimensional scaling plots showing the effect of primer set on unweighted and weighted UniFrac distances.** Data underlying this figure can be found in S2 and S3 Data.
(TIF)

**S9 Fig. Nonmetric multidimensional scaling plots showing the effect of sequencing depth on unweighted and weighted UniFrac distances.** Low depth is <20,000 reads, medium depth is 20,000–49,999 reads, and high depth is ≥50,000 reads. Data underlying this figure can be found in S2 and S3 Data.
(TIF)

**S10 Fig. Barplots of observed and expected numbers of individuals by age category, sex, delivery mode, and infant diet for each racial category.** Data underlying this figure can be found in S5 Table.
(EPS)

**S11 Fig. Barplots of observed and expected numbers of individuals by age category, sex, delivery mode, and infant diet for each ethnicity category.** Data underlying this figure can be found in S5 Table.
(EPS)

**S12 Fig. Nonmetric multidimensional scaling plots showing the effect of race on weighted UniFrac distances for additional age categories.** Data underlying this figure can be found in S2 and S3 Data.
(EPS)

**S13 Fig. Nonmetric multidimensional scaling plots showing the effect of race on unweighted UniFrac distances within age categories.** Data underlying this figure can be found in S2 and S3 Data.
(EPS)

**S14 Fig. Correlation plots showing the association between age and species relative abundance by race.** Taxa that are differentially abundant across age categories according to ANCOM-BC results and were identified as differentially abundant between racial categories are included. Data underlying this figure can be found in S8 Data.
(EPS)

**S15 Fig. Correlation plots showing the association between age and species relative abundance by ethnicity.** Taxa that are differentially abundant across age categories according to ANCOM-BC results and were identified as differentially abundant between ethnicity categories are included. Data underlying this figure can be found in S9 Data.
(EPS)

**S16 Fig. Feature importance from a random forest model used to identify taxa distinguishing children of different self-identified racial categories.** Dots denote the median importance, and whiskers denote 95% confidence intervals. Data underlying this figure can be found in S10 Data.
(TIFF)

**S17 Fig.** Relative abundances across White (blue), Black (yellow), and Asian/Pacific Islander (red) children of the 13 taxa identified as (1) important features in the random forest model; (2) differentially abundant in the ANCOM analysis; and (3) differentially abundant in a previous study of adult gut microbiomes. All boxplots show the median and interquartile range (IQR), and whiskers extend to 1.5*IQR. Relative abundances for boxplots are square root transformed. Data underlying this figure can be found in S11 Data.
(EPS)

**S18 Fig. Receiver operating characteristic (ROC) curves for a random forest model classifying adult gut microbiome samples by race using samples from children as a training dataset.** Shading represents a 50% confidence interval around the median. Data underlying this figure can be found in S12 Data.
(TIFF)

**S19 Fig. Feature importance from a random forest model used to identify taxa distinguishing adults of different self-identified racial categories based on data from children.** Dots denote the median importance, and whiskers denote 95% confidence intervals. Data underlying this figure can be found in S13 Data.
(TIFF)

**S20 Fig. Relative abundance of highly important features from the random forest models using data from multiple child microbiome studies and adults from the American Gut Project.** Enterobacteriaceae and *Prevotella* (A and B) were highly important in the child–child models and *Ruminococcus* (C) was highly important in the child–adult models. All boxplots show the median and interquartile range (IQR), and whiskers extend to 1.5*IQR. Relative abundances for boxplots are square root transformed. Data underlying this figure can be found in S14 Data.
(EPS)

**S21 Fig. Box plots showing sequencing depth (number of forward reads prior to filtering for each sample) by study.** Data underlying this figure can be found in S15 Data.
(EPS)

**S22 Fig. Box plots showing Shannon diversity and observed ASV alpha diversity metrics by age.** Data underlying this figure can be found in S1 Data.
(TIF)

**S23 Fig. Box plots showing Shannon diversity and observed ASV alpha diversity metrics by race.** Data underlying this figure can be found in S1 Data.
(TIF)

**S24 Fig. Box plots showing Shannon diversity and observed ASV alpha diversity metrics by ethnicity.** Data underlying this figure can be found in S1 Data.
(TIF)

**S25 Fig. Box plots showing Shannon diversity and observed ASV alpha diversity metrics by sex.** Data underlying this figure can be found in S1 Data.
(TIF)

**S26 Fig. Box plots showing Shannon diversity and observed ASV alpha diversity metrics by infant diet.** Data underlying this figure can be found in S1 Data.
(TIF)

**S27 Fig. Box plots showing Shannon diversity and observed ASV alpha diversity metrics by delivery mode.** Data underlying this figure can be found in S1 Data.
(TIF)

**S1 Table. Characteristics of studies included in the analysis.**
(XLSX)

**S2 Table. Permutational multivariate analysis of variance (PERMANOVA) and homogeneity of variance (Beta dispersion) test statistics.**
(XLSX)

**S3 Table. Pairwise PERMANOVAs statistics for race, ethnicity, and study in the full dataset, as well as race, ethnicity, age, sex, delivery mode, and infant diet for samples where all variables were available.**
(XLSX)

**S4 Table. Linear mixed effects model statistics for alpha diversity comparisons.** Model statistics are reported on the table on the left, and pairwise comparison statistics are presented in the table on the right for variables that were significant.
(XLSX)

**S5 Table. Observed vs. expected numbers of samples for each metadata variable of interest between race and ethnicity categories.**
(XLSX)

**S6 Table. Test statistics for differential abundance analyses at the phyla level.**
(XLSX)

**S7 Table. Test statistics for differential abundance analyses at the family level.**
(XLSX)

**S8 Table. Test statistics for differential abundance analyses at the genus level.**
(XLSX)

**S9 Table. Test statistics for differential abundance analyses at the species level.**
(XLSX)

**S10 Table. Genera identified as differentially abundant between self-identified racial categories across studies.**
(XLSX)

**S11 Table. Important features identified with the random forest classifiers.** Both child–child and child–adult models are listed.
(XLSX)

**S1 Data. Alpha diversity values (Faith's PD, Observed features, Shannon diversity, Pielou's evenness, Chao1) for all samples along with metadata shown in Figs 1A and S22–S27.**
(XLSX)

**S2 Data. MDS1 and MDS2 values for weighted UniFrac distances along with metadata shown in Figs 1B, 2B, S1–S9, and S12.**
(XLSX)

**S3 Data. MDS1 and MDS2 values for weighted UniFrac distances along with metadata shown in Figs S1–S9 and S13.**
(XLSX)

**S4 Data. Confidence intervals for Tukey contrasts from linear mixed effects models of the effect of race and ethnicity on alpha diversity for the 0–2.9 month, 3–11.9 month, and 12–35.9 month age categories.** Tukey contrasts were performed using the multcomp package in R after running linear mixed effects models using the lme4 package in R. The values below are from the summary output of those contrasts.
(XLSX)

**S5 Data. Relative abundance of taxa plotted in Fig 3A along with race.**
(XLSX)

**S6 Data. Taxa that are differentially abundant in children (this study) and adults [3].**
(XLSX)

**S7 Data. Sensitivity, specificity, and false positive rates output from the child-only random forest model.** These data were used to construct the ROC curve in Fig 3C.
(XLSX)

**S8 Data. Relative abundance of taxa in S14 Fig along with race and age metadata.**
(XLSX)

**S9 Data. Relative abundance of taxa in S14 Fig along with ethnicity and age metadata.**
(XLSX)

**S10 Data. Feature importance values for the child-only random forest model.**
(XLSX)

**S11 Data. Relative abundance of taxa plotted in S17 Fig along with race.**
(XLSX)

**S12 Data. Sensitivity, specificity, and false positive rates output from the child-only random forest model.** These data were used to construct the ROC curve in S18 Fig.
(XLSX)

**S13 Data. Feature importance values for the child-adult random forest model.**
(XLSX)

**S14 Data. Relative abundance of taxa plotted in S21 Fig along with race and age group (adults or children).**
(XLSX)

**S15 Data. Sequencing depth for all samples included in the analysis along with study.**
(XLSX)

## Acknowledgments

Computational resources were supported by the Vanderbilt Microbiome Innovation Center. This work was conducted in part using the resources of the Advanced Computing Center for Research and Education (ACCRE) at Vanderbilt University.

## Author Contributions

**Conceptualization:** Elizabeth K. Mallott, Seth R. Bordenstein.

**Data curation:** Alexandra R. Sitarik, Leslie D. Leve, Camille Cioffi, Carlos A. Camargo, Jr, Kohei Hasegawa.

**Formal analysis:** Elizabeth K. Mallott.

**Funding acquisition:** Alexandra R. Sitarik, Leslie D. Leve, Camille Cioffi, Carlos A. Camargo, Jr, Kohei Hasegawa, Seth R. Bordenstein.

**Investigation:** Alexandra R. Sitarik, Leslie D. Leve, Camille Cioffi, Carlos A. Camargo, Jr, Kohei Hasegawa.

**Methodology:** Elizabeth K. Mallott, Alexandra R. Sitarik, Leslie D. Leve, Camille Cioffi, Carlos A. Camargo, Jr, Kohei Hasegawa.

**Supervision:** Seth R. Bordenstein.

**Visualization:** Elizabeth K. Mallott.

**Writing – original draft:** Elizabeth K. Mallott.

**Writing – review & editing:** Elizabeth K. Mallott, Alexandra R. Sitarik, Leslie D. Leve, Camille Cioffi, Carlos A. Camargo, Jr, Kohei Hasegawa, Seth R. Bordenstein.

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
