## [Editor Report · Decision Letter 0]

2 Nov 2022

Dear Dr. Mallott, 

Thank you for submitting your manuscript entitled "Sociodemographic-linked microbiome variation appears after three months of age" for consideration as a Research Article by PLOS Biology.

Your manuscript has now been evaluated by the PLOS Biology editorial staff, as well as by an academic editor with relevant expertise, and I am writing to let you know that we would like to send your submission out for external peer review as a Short Report. Please select Short Report where corresponds when re-submitting your manuscript. 

PLOS Biology Short Reports are research articles that may be preliminary, based on a small number of experiments that might not completely flesh out the biological phenomenon under study. However, we expect their conclusions to be fully supported by the data. Equally importantly, we aim for our Short Reports to be provocative and of general interest, in such a way as to spur future research.

Once your full submission is complete, your paper will undergo a series of checks in preparation for peer review. After your manuscript has passed the checks it will be sent out for review. To provide the metadata for your submission, please Login to Editorial Manager (https://www.editorialmanager.com/pbiology) within two working days, i.e. by Nov 04 2022 11:59PM.

Kind regards,

Paula

---

Senior Editor

PLOS Biology

---

## [Decision Letter · Decision Letter 1]

9 Feb 2023

Dear Dr. Mallott,

Please allow me to first apologize for the delay in the processing of your manuscript. This delay is caused by my difficulty in recruiting reviewers for your manuscript, and is further compounded by one referee promising an overdue report but failing to deliver after long delay and multiple chases. I am sorry for this unexpected event. Thank you for your patience while your manuscript "Sociodemographic-linked microbiome variation appears after three months of age" was peer-reviewed at PLOS Biology. It has now been evaluated by the PLOS Biology editors, an Academic Editor with relevant expertise, and by several independent reviewers. 

In light of the reviews, which you will find at the end of this email, we would like to invite you to revise the work to thoroughly address the reviewers' reports. As you will see below, all the reviewers find the manuscript interesting but raise some important issues that will need to be solved before publication. Please address all the reviewers' comments. 

Given the extent of revision needed, we cannot make a decision about publication until we have seen the revised manuscript and your response to the reviewers' comments. Your revised manuscript is likely to be sent for further evaluation by all or a subset of the reviewers.

**IMPORTANT - SUBMITTING YOUR REVISION**

*Re-submission Checklist*

*Published Peer Review*

*PLOS Data Policy*

*Blot and Gel Data Policy*

Sincerely,

Paula

---

Senior Editor

PLOS Biology

REVIEWS:

Reviewer #1: Gut microbiome

Reviewer #2: Gut microbiome

Reviewer #1: The authors present a meta-analysis of existing datasets with the thesis that differences exist in the gut microbial communities of children driven by factors captured either by race self-identified race and ethnicity or socio economic status (SES). While the authors would have preferred to use comprehensive measurements of SES, this was not possible due to lack of available metadata from the chosen manuscripts. This problem is common, and the authors instead used available data, and to their credit identify signals connecting race/ethnicity to differences in the gut microbiome. There is some literature on this that the author's appropriately reference, though this literature base needs to continue to grow.

The authors major thesis is that differences in the gut microbial communities of children with different self-identified race and ethnicity features emerges after the age of 3 months. The authors are capable of completing a comprehensive analyses of the gut microbiome, but major issues, which they can address, need to be considered prior to publication. It is conceivable that the short report format may be too restrictive for these authors and if the editors are amenable a longer format be made available to the authors.

Major Critiques:

- A core thesis of this work is that differences in race/ethnicity interact with age. For this particular manuscript, this requires a direct assessment of the impact of Age on the gut microbiome at the taxonomic level is required. There is a superficial portion of this analysis already completed (see Figure 1A, Supplemental Table 10). Analysis of differentially abundant taxa that vary with age with plots of those taxa with age are critical. For example, do all race and ethnic groups start with similar levels of Bifidobacterium (linked to breast feeding) or Lactobacillus? Are the species the same? Are there differences at the genus level? Shannon and beta diversity are key similarities at early ages the authors describe, though specific species level similarities need to be described as well. Plots of taxa that correlate with age could be plotted in the same figure where taxa that are different at later ages are plotted against age by race/ethnicity

- The authors do not account for sequencing technology of the original studies. This should be taken into account (Illumina vs other; base pair size) in addition to depth of coverage and type of sequencing (metagenomic vs 16S). These variables should be incorporated and their contributions to variations explained in Figure S6. In addition sampling depth of each study should be plotted and this data made available.

- What is the relationship between race/ethnicity and both the variables mode of delivery and breast feeding for data where these data are available. This is a crucial piece of information that requires visualization in this data set. For example, if breast feeding and mode of delivery are very different in one of the analyzed groups, that would be a key thing to highlight. This could reflect the biologic basis of differences between groups detected later on. This data is available in some of the tables (S11) though needs to be visually represented in a figure.

Minor issues

- Line 74-75 "As age had the strongest association…" describe how you came to this conclusion. It looks like Subject had the greatest explained variation (Figure S2) followed by Age. Subject would be predicted to explain the greatest variation, so the conclusions that Age was the next largest variable (and largest for the logical argument of the authors) is reasonable but the rationale needs to be more explicit. I don't disagree with the interpretation but the reader will benefit a lot from understanding the logic here.

- Use different symbols in Figure 2B as it is not clear. Try letters or open and closed shapes. These shapes are very difficult to discern at the size presented in the MS.

- Table formatting is very limited. While the structure of the formatting is OK, borders, headers, shading all need to be added to make these tables easier to review. There is a lot of good information here, and it is a testament to the authors they include this material, but it needs to be cleaned up heavily for formal publication.

- Strata was used for PERMANOVA analysis. This is a fair approach. The authors should list an example of the specific equation in their methods

- The random forest methodology appears sound, though the method text should be clarified. What was repeated 100 times? Can the authors use the data from one study to predict the sample race/ethnicity of another study? If this cannot be done, it is only a minor issue, but if can be done it would add considerably to the manuscript. 

Reviewer #2: Mallot et al. provide a meta-analysis of 16S gut microbiome composition datasets for humans under 3 years old across racial and ethnic categories to show that differences arise shortly after birth between these groups. Machine learning is used to show that thse differences can be used to classify the groups and the features important in these models correspond to those used to classify adult identity. Together the data are consistent with the differences previously described in adults arising early in life.

Major comments

The presentation and discussion of the data seems biased. The authors should be careful in mis-interpretation or over-interpretation, particularly in connection to disease, since this may be counterproductive in the long run for this important field. There is discussion and references in the taxonomic differences results section (starting at line 114) with regard to various taxa that are enriched or depleted and their associations with allergy, etc consistent with the authors' premise. However, earlier in the manuscript the authors show that Black children have higher alpha diversity, yet the abundant literature that associates alpha diversity with health and low diversity (seen in the white children here) with many inflammatory diseases is not discussed or referenced. Therefore, the data appears to be sending a mixed signal that the authors gloss over.

Another example is in lines 161-4 and figure S14: 

"In contrast, Ruminococcus is specifically important in the child-adult models, likely due to similar variation in abundance between racial categories in both children and adults (Figure S14). Higher abundances of Ruminococcus are linked with an increased risk of colorectal cancer (37), a disease for which there is a known racial health disparity (38).

One would expect based on this statement that Ruminococcus is more abundant in Black and Asian/Pacific Islander, however in S14 it appears to be more abundant in white for both children and adults. If this taxon's contribution is due to enrichment in whites, why do the authors point out its association with diseases that are associated with health disparities. The data appears to support the opposite, so please clarify.

Another important point is that the authors appear to neglect the major taxonomic shift that is known to occur in infants upon the introduction of solid food. The final part of the paper that examines differences in taxa is performed across all ages. Since the infant microbiome from 0-3 years can be thought of as existing in two states, with some transition between them -- pre-solid food and post-solid food. While the timing of solid food introduction may not be available, the authors may use other criteria to create these two bins, such as their initial analysis, changes in abundance of key taxa known to change developmentally, or an age that would generally make sense as a cut-off. You could imagine 0-6 months as pre-solid food, 6-12 months as a transition (maybe don't use this data) and >12 months as post-solid food. 

Breaking down this analysis is important for multiple reasons since the initial beta-diversity analysis indicates that the significant differences appear at later ages. This separation of early and late may also enable detection of early life taxonomic differences that currently go undetected due to the larger effect and noise introduced from the data of older children.

Specific points

If publications exists where these datasets were first reported, please include in table S1. Also include the author identifiers (eg "Kim") in the columns so this can help track the datasets between figures and the table.

The main message of the supplemental figs S1-S6 appears to be that study appears to correspond to the separation of the two clusters in weighted and unweighted (S6). Since there is an uneven distribution of data across studies, particularly at the extreme young and old ages (table S1), the authors should explain how this was accounted for in the analysis to avoid study-specific artifacts.

Why are the colors different between the weighted and unweighted in S6? For example, there is a lot of brown in the left graph and a lot of green in the right graph. Are these not the same data from the studies plotted in two different ways?

L 85. "Pairwise comparisons

86 confirmed that Black individuals had higher within-sample diversity than White individuals at 3-

87 11.9 months and 1-2.9 years". 

Would be good to add "for at least one of the five measures of diversity" to recognize this doesn't reflect all measures.

Fig. S7-S8. 

-Why are there more age bins created for the unweighted (S8) vs. the weighted (S7)?

-The groupings in the weighted appear to match the study specific effects. For example, the collapse of the two clusters to one main cluster in the last two groups may correspond to the oldest group having data that is dominated by one study. So are the two groupings in the younger plots (2.9 years and younger) reflective of study-specific differences or race and ethnicity? Perhaps there is a better way to present the data? 

Fig. 2 legend should be reorganized so the text associated with each panel is separated. Currently the information for A and B is intertwined and it is a bit confusing.

"0-2.9 months of age, 3-11.9 months of age, and 1-2.9 years of age". The discontinuity of numbers and units is a bit awkward. Recommend keeping everything in months and making the last category 12-35.9 throughout the figures and text.

To identify differentially abundant taxa by race and ethnicity, the authors perform an analysis across all age categories. This is perhaps not so intuitive for the reader considering the previous section showing that younger ages do not have significant differences. To avoid skepticism of post-hoc decisions, the authors should either clearly articulate rationale for why they included the younger ages or present supplemental data showing how exclusion of the younger ages impacts the results.

L 121. "Four of the 19 overlapping taxa were higher in abundance in both Black children and

122 adults compared with White children and adults, and four of the overlapping taxa were lower in

123 abundance in both Black children and adults."

What about the 11 overlapping taxa that are not mentioned. If 8 show similar enrichment patterns, but 11 show opposite patterns, what does this say about the concordance between analyses/studies? Please clarify how this is different from random. 

L 126. "these taxa have

127 been associated with an increased risk of autoimmune and allergic diseases, asthma, and obesity

128 across human populations (28-32)." Should be "across industrialized human populations". This is important to note since so many non-industrialized populations are enriched in taxa such as Prevotella copri and have low incidence of autoimmune and allergic disease.

L 131. "Conversely, Veillonella, which decreases the risk of asthma and allergic

132 disease(28,33)," Careful with wording since as stated, a causal relationship is implied, yet the references appear to only test associations. Maybe, "which is associated with decreased risk of".

L 157. "However, the taxa with the highest importance differed with

158 respect to the magnitude and direction of the differences between adults and children". I think this means that the enrichment of taxon X can help predict race Y in children, yet depletion of taxon X can predict race Y in adults. If this is the case, it brings up the question of what this means biologically. It would be good for the authors to discuss cases, and provide some specific examples, where taxa are important in both cases, but show different directions in adults and children.

---

## [Decision Letter · Decision Letter 2]

14 Jun 2023

Dear Dr. Mallott,

Thank you for your patience while we considered your revised manuscript "Sociodemographic-linked microbiome variation appears after three months of age" for publication as a Short Reports at PLOS Biology. This revised version of your manuscript has been evaluated by the PLOS Biology editors, the Academic Editor and one of the original reviewers.

Based on the reviews and on our Academic Editor's assessment of your revision, we are likely to accept this manuscript for publication, provided you satisfactorily address the following data and other policy-related requests.

1. DATA POLICY:

A) Supplementary files (e.g., excel). Please ensure that all data files are uploaded as 'Supporting Information' and are invariably referred to (in the manuscript, figure legends, and the Description field when uploading your files) using the following format verbatim: S1 Data, S2 Data, etc. Multiple panels of a single or even several figures can be included as multiple sheets in one excel file that is saved using exactly the following convention: S1_Data.xlsx (using an underscore).

B) Deposition in a publicly available repository. Please also provide the accession code or a reviewer link so that we may view your data before publication.

Regardless of the method selected, please ensure that you provide the individual numerical values that underlie the summary data displayed in the following figure panels as they are essential for readers to assess your analysis and to reproduce it: Figures 1AB, 2AB, 3ABC, Supplementary Figures S1AB, S2, S3AB, S4AB, S5AB, S6AB, S7AB, S8AB, S9AB, S10, S11, S12, S13, S14, S15, S16, S17, S18, S19, S20ABC, S21, S22ABC, S23ABC, S24ABC, S25ABC, S26ABC, S27ABC.

**Please also ensure that figure legends in your manuscript include information on where the underlying data can be found, and ensure your supplemental data file/s has a legend.**

2. Does the provided code reproduce all of the papers' results (or can it by the time of publication)? Is or will there be instructions on how to do that? Please note that sole deposition of data or code to GitHub would not be compliant with our policies, as this could be changed after publication (https://journals.plos.org/plosbiology/s/data-availability). However, once the data/code is final, you can archive your publicly available GitHub data to Zenodo. Once you do this, it will also generate a DOI number that you can provide us with. See the process for doing this here: https://docs.github.com/en/repositories/archiving-a-github-repository/referencing-and-citingcontent

3. We suggest a change in the title: "Human microbiome variation associated with race and ethnicity emerges as early as three months of age".

We expect to receive your revised manuscript within two weeks.

*Published Peer Review History*

*Press*

Sincerely,

Paula

---

Senior Editor,

pjaureguionieva@plos.org,

PLOS Biology

Reviewer remarks:

Reviewer #1: Vaibhav Upadhyay

This is a wonderful study in an area of major importance. The authors have appropriately responded to my revisions and conducted a wonderful meta-analysis. I commend them on their work and wish them luck pursuing further research in this area of great interest to the broader scientific community. Congrats.

---

## [Editor Report · Decision Letter 3]

3 Jul 2023

Dear Dr Mallott,

Thank you for the submission of your revised Short Reports "Human microbiome variation associated with race and ethnicity emerges as early as three months of age" for publication in PLOS Biology. On behalf of my colleagues and the Academic Editor, Jotham Suez, I am pleased to say that we can in principle accept your manuscript for publication, provided you address any remaining formatting and reporting issues. These will be detailed in an email you should receive within 2-3 business days from our colleagues in the journal operations team; no action is required from you until then. Please note that we will not be able to formally accept your manuscript and schedule it for publication until you have completed any requested changes.

PRESS

Sincerely, 

Paula 

---

Senior Editor

PLOS Biology
